# Structural and Thermodynamic Peculiarities of Core-Shell Particles at Fluid Interfaces from Triangular Lattice Models

**DOI:** 10.3390/e22111215

**Published:** 2020-10-26

**Authors:** Vera Grishina, Vyacheslav Vikhrenko, Alina Ciach

**Affiliations:** 1Department of Mechanics and Engineering, Belarusian State Technological University, 13a Sverdlova Str., 220006 Minsk, Belarus; vera1grishina@gmail.com (V.G.); vvikhre@gmail.com (V.V.); 2Institute of Physical Chemistry, Polish Academy of Sciences, Kasprzaka 44/52, 01-224 Warszawa, Poland

**Keywords:** core-shell particles, liquid interfaces, triangular lattice, thermodynamics, ground states, structure, line tension, phase coexistence, competing interaction, fluctuations

## Abstract

A triangular lattice model for pattern formation by core-shell particles at fluid interfaces is introduced and studied for the particle to core diameter ratio equal to 3. Repulsion for overlapping shells and attraction at larger distances due to capillary forces are assumed. Ground states and thermodynamic properties are determined analytically and by Monte Carlo simulations for soft outer- and stiffer inner shells, with different decay rates of the interparticle repulsion. We find that thermodynamic properties are qualitatively the same for slow and for fast decay of the repulsive potential, but the ordered phases are stable for temperature ranges, depending strongly on the shape of the repulsive potential. More importantly, there are two types of patterns formed for fixed chemical potential—one for a slow and another one for a fast decay of the repulsion at small distances. In the first case, two different patterns—for example clusters or stripes—occur with the same probability for some range of the chemical potential. For a fixed concentration, an interface is formed between two ordered phases with the closest concentration, and the surface tension takes the same value for all stable interfaces. In the case of degeneracy, a stable interface cannot be formed for one out of four combinations of the coexisting phases, because of a larger surface tension. Our results show that by tuning the architecture of a thick polymeric shell, many different patterns can be obtained for a sufficiently low temperature.

## 1. Introduction

Metal or semiconducting nanoparticles find numerous applications in catalysis, optics, biomedicine, environmental science, and so forth. In order to prevent the charge-neutral nanoparticles from aggregation, recently, various types of core-shell nanoparticles (CSNPs) have been produced [1,2]. In the CSNPs, the hard, typically metal or semiconducting nanoparticle with a diameter ranging from a few tens to a few hundreds of nanometers is covered by a soft polymeric shell. The polymeric chains can interpenetrate, and the distance between the particles can become smaller than the shell diameter at some energetic cost. This energetic cost, or the softness of the shells, can be controlled in particular by the crosslinking of the polymeric chains. The shell-to-core diameter ratio in the majority of the experiments varies from about 1.1 to about 4 [2,3,4,5]. Since the shell thickness can be controlled independently of the core diameter, and the effective interactions between the CSNPs depend on the thickness and the architecture of the shells, the desired effective interactions can be obtained by choosing different protocols for the synthesis.

Properties of the CSNPs have been intensively studied not only in the bulk, but also at the fluid interfaces [1,2,3,4,5,6,7,8,9,10,11,12]. It turned out that while in the bulk, the effective interaction consists of the hard core followed by the soft repulsive shoulder; at fluid interfaces, strong capillary attraction can be present for separations larger than the shell diameter, in addition to the above mentioned repulsive interactions at shorter distances [2,3,4].

Experimental results show that the CSNPs at fluid interfaces can form interesting patterns, depending on the properties of the particles and on the fraction of the interface area covered by them [2,3,4]. For a small area fraction, highly ordered arrays of hexagonally packed particles are typically observed. Compression may lead to the sudden formation of particle clusters [1]. The surface pressure–area isotherms can have a characteristic shape of alternating segments with a very large and quite small slope, and the large compressibility signals structural changes [2,3]. The origin of these patterns and of the structural changes, their nature, and dependence on the properties of the CSNPs are not fully understood yet. The theory of CSNPs, that at fluid interfaces, they repel each other at short separations and attract each other at large separations, is much less developed than the experimental studies [13,14,15]. This is in contrast to theoretical and simulation studies of patterns formed by particles with soft repulsive potentials [16].

Because there are many factors controlling the core-and-shell diameter and the architecture of the shell, there is a need for a simplified, coarse-grained theory that could predict general trends in pattern formation for various ranges, strengths, and shapes of the effective potential. In Ref. [13], a one-dimensional (1D) lattice model with repulsion between nearest neighbors and attraction between second or third neighbors was solved exactly. The obtained isotherms consist of alternating segments with very large and quite small slopes, as in experiments. The steep parts of the isotherms are associated with periodic patterns. The number of the steps, however, is larger in the case of third-neighbor attraction (i.e., thicker shell), and depends on the strengths of the repulsive and attractive parts of the potential. There are no phase transitions in the thermodynamic sense in 1D, but the correlation function shows oscillatory decay with the correlation length that for a strong attraction can be very large ($10^{4}$ times the core diameter). These results show a strong dependence of the structure and mechanical properties of the monolayers of the CSNPs on the range, shape, and strength of the effective interactions and agree with the experimental observation of the more complex behavior of the CSNPs with thick shells. However, any 1D model cannot answer the question if different patterns correspond to different phases, and obviously only 1D patterns can be examined.

Particles with a size equal to or larger than a few tens of nanometers are practically irreversibly adsorbed at the interface, but can move freely in the interface area [2]. For this reason, the particles trapped at the interface can be modeled as a two-dimensional system. Since closely packed CSNPs form a hexagonal pattern, triangular lattice models with the lattice constant *a* equal to the diameter of the hard core (or the distance of the closest approach of the particles upon compression) are appropriate and convenient generic models for CSNPs at fluid interfaces. In Ref. [15], lattice models for CSNPs with thin and thick shells were introduced and studied. Following Ref. [16], we assumed that the shell-to-core ratio separating the thin and thick shells is $\sqrt{3}$. According to this criterion, the shell is thin when the shells of the second neighbors of closely-packed particle-cores do not overlap, otherwise it is thick.

For thin shells, nearest-neighbor repulsion and second neighbor attraction between particle cores occupying sites of the triangular lattice were assumed. The shell-to-core ratio in this model is $\sqrt{3}$. We have found four phases in this model—very dilute gas, hexagonal lattice of closely packed shells, hexagonal lattice of vacancies, and closely packed cores. We have calculated the surface tension between coexisting phases for different orientations of the interface and found that the particles at the stable interfaces corresponding to the smallest surface tension lie on straight lines. The interface lines meet at the angles 60° or 120°. When the fixed area fraction of the CSNPs is smaller than the area fraction of the hexagonal phase, large voids with a hexagonal shape are formed. The results are in good agreement with the experiment in Reference [2,3,4].

In order to study the effect of the shell thickness, in the second model, we assume that the inner shell is covered by a much softer outer shell, and the nearest-neighbor repulsion is followed by vanishing interactions for the second, third, and fourth neighbors and by attraction between the fifth neighbors. The shell-to-core ratio in this model is equal to 3, as in the experiments of Refs. [1,3]. Six more phases, including honeycomb lattices of particles or vacancies and periodically ordered rough clusters were found, but these additional phases were stable only at the coexistence with the phases found for the thin-shell model. For the fixed area fraction of the particles, two phases with the closest area fraction to the mean one, and the interface between them, were present at low temperature. For increasing *T*, islands of different phases in the sea of the hexagonally packed shells were observed in the course of simulations for the area fraction exceeding the value at the close packing of the shells. Such complex patterns, somewhat similar to the patterns observed in experiments [2,3,4], occur because of the metastability of several ordered phases and large interface fluctuations in 2D.

The results of Reference [15] concern CSNPs with composite shells with a stiff inner shell and very soft outer shell. In this work, we focus on the question of the role of the shape of the repulsive shoulder, associated with the architecture of the crosslinked polymeric chains. We assume first- and second-neighbor repulsion, and fifth-neighbor attraction, and consider different second- to first-neighbor repulsion ratios. The model is introduced in Section 2. The ground state of an open system and of the system with a fixed number of particles is determined in Section 3.1 and Section 3.2, respectively. We find the same patterns as in Ref. [15] for weak second-neighbor repulsion, but the patterns absent for thin shells are present for some intervals of the chemical potential. For the second-to-first neighbor repulsion ratio larger than 1/3, the stable patterns are completely different. Moreover, for some ranges of the chemical potential, the ground state is degenerated, and two quite different patterns are stable. In Section 4, thermodynamic properties obtained for T>0 by Monte Carlo simulations are described. We present the chemical potential, compressibility, and specific heat as functions of the concentration. Section 5 contains our conclusions.

## 2. The Model

The system that models the core-shall particles on a surface is described in Ref. [15]. The thermodynamic Hamiltonian of the system is:(1)$$H^{*}=\frac{1}{2}\sum_{k=1}^{k_{max}}\sum_{k_{i}=1}^{z_{k}}\sum_{i=1}^{M}J_{k}^{*}{\hat{n}}_{i}{\hat{n}}_{k_{i}}-\mu^{*}\sum_{i=1}^{M}{\hat{n}}_{i},$$
where $k_{i}$ numerates the sites of the *k*-th coordination sphere around the site *i*, $z_{k}$ is the coordination number, $J_{k}^{*}$ is the interaction constant for the *k*-th coordination sphere, ${\hat{n}}_{i}$ is the occupation number (0 or 1), and $\mu^{*}$ is the chemical potential. The particles can occupy sites of a triangular lattice containing M=L×L lattice sites.

The lattice parameter *a* is equal to the diameter of the hard core of the particles. It is supposed that the particles repel each other with the intensity $J_{1}^{*}=J_{1}J$ ($J_{1}=1$), if the particles occupy the nearest neighbor sites, and feel weaker repulsion on the next nearest sites with the intensity $J_{2}^{*}=J_{2}J$. The intermediate third and fourth neighbors do not interact ($J_{3}=J_{4}=0$), while the fifth neighbors attract each other with the energy $J_{5}^{*}=-J_{5}J$. Thus, $J_{2}$ and $J_{5}$ are the dimensionless interaction energies ($J_{2}=J_{2}^{*}/J$, $J_{5}=-J_{5}^{*}/J$, and *J* has units of energy). The dimensionless chemical potential $\mu =\mu^{*}/J$ and dimensionless temperature $T=k_{B}T^{*}/J$ will be used as well. In the terminology of Reference [15], it is model II with an additional repulsive interaction of the second neighbors.

The interaction potential as a function of the distance between the particle cores is shown in Figure 1. As is demonstrated below, the variation of the second and fifth neighbor interactions can lead to different symmetry-breaking (heterostructural) transitions in the system; it was recently attained using the augmented potential [17]. Compared to conventional interaction potentials that are determined a priori, the augmented potentials adjust the effective interactions on the basis of the local environment of each particle and efficiently capture multi-body effects at a local level.

In accordance with the range of the interparticle interactions (up to the fifth neighbors), the unit cell contains nine (3×3) lattice sites. In this case, the fifth neighbors (that correspond to the largest interaction range) belong to the nearest unit cells and the distance between them determines the translation vector that preserves the symmetry of the system. The subsequent calculation (Section 3) and simulation (Section 4) shows that such a choice accounts for all possible ordered states of the system. For describing the ordered states, the lattice is decomposed in nine sublattices (Figure 2).

## 3. The Ground States

### 3.1. The Ground States of Open Systems

In the system with repulsive interactions of the first and second neighbors and attraction of the fifth neighbors, the ground states with ten concentrations n/9, n=0,1,...,9 with a different distribution of the particles over the unit cell are possible. At zero temperature, the stable configurations are determined by the minima of the dimensionless thermodynamic Hamiltonian per lattice site
(2)$$\omega =H^{*}/MJ$$
because the entropy does not contribute to the thermodynamic functions.

In the vacuum state $\omega_{0}=0$. In the c=1/9 phase, each particle has six neighbors of the fifth order. Calculating the system energy, each interacting bond is taken into account twice. Thus,
(3)$$\omega_{1/9}=\left(-3J_{5}-\mu \right)/9.$$

For c=2/9, two possibilities exist for the distribution of two particles over the unit cell (Figure 3). The calculated potentials are as follows:(4)$$\begin{array}{cc} \; & \left(a\right)\;\omega_{2/9}=\left(3J_{2}-6J_{5}-2\mu \right)/9,\\ \; & \left(c\right)\;\omega_{2/9}=\left(1-6J_{5}-2\mu \right)/9. \end{array}$$
the 1 in Equation (Equation 4)c, originates from the nearest neighbor interaction for this configuration.

For larger concentrations, we can write the expressions corresponding to the columns a–c in Figure 3
(5)$$\begin{array}{cc} \; & \left(a\right)\;\omega_{3/9}=\left(9J_{2}-9J_{5}-3\mu \right)/9,\\ \; & \left(b\right)\;\omega_{3/9}=\left(2+3J_{2}-9J_{5}-3\mu \right)/9,\\ \; & \left(c\right)\;\omega_{3/9}=\left(3-9J_{5}-3\mu \right)/9, \end{array}$$
(6)$$\begin{array}{cc} \; & \left(a\right)\;\omega_{4/9}=\left(3+9J_{2}-12J_{5}-4\mu \right)/9,\\ \; & \left(b\right)\;\omega_{4/9}=\left(4+6J_{2}-12J_{5}-4\mu \right)/9,\\ \; & \left(c\right)\;\omega_{4/9}=\left(5+3J_{2}-12J_{5}-4\mu \right)/9. \end{array}$$

For the concentration c=n/9 with n≥5, the distribution of the vacancies in the unit cell is the same as the distribution of the particles for c=(9−n)/9. In the dense state (n=9, c=1), all the lattice sites are filled by the particles. The ω for these states can be calculated as
(7)$$\omega_{n/9}=\omega_{1-n/9}+\left(2n-9\right)\left[3\left(1+J_{2}-J_{5}\right)-\mu \right]/9,\;5\leq n\leq 9$$
for each particular distribution of the particles/vacancies over the unit cell.

Comparing the r.h.s. of Equations (Equation 4)–(Equation 6), we can see that at $J_{2}<1/3$ the system states shown in the column (a) of Figure 3 are more stable, while at $J_{2}>1/3$, the system states shown in the column (c) are preferable. The system states of the column (b) could occur if additional interactions of the third and/or fourth neighbors were taken into account.

Thus, the presence of the interaction between the second neighbors eliminates the phase degeneration observed in the system without it [15] and results in additional stable phases in certain regions of the chemical potential.

To make further analysis more transparent, we consider the case $J_{5}=J_{2}\equiv J_{2,5}$. There exists the crossover value of the interaction parameter $J_{2,5}=1/3$, which separates the possible states of the system. Let us consider particular values of the interaction parameter, below and above the crossover, $J_{2,5}=1/4<1/3$ and $J_{2,5}=1/2>1/3$, that correspond to the slower and faster decay of the interaction potential for short separations as compared with V(r), shown in Figure 1, respectively.

In Figure 4, the grand thermodynamic potentials per lattice site, Equations (Equation 3)–(Equation 7), are shown as functions of the chemical potential at $J_{2,5}=1/4$. The stable states correspond to the lowest value of ω for given μ, i.e., to the lowest line segments between the intersection points. These segments determine the chemical potential intervals corresponding to particular concentrations of the particles. Each intersection point corresponds to the coexistence of two phases with the closest concentrations. The stable system states and the corresponding chemical potential intervals are shown in Figure 5. The phase diagram of the system at $J_{2,5}<1/3$ is shown in Figure 6.

The phase coexistence lines between the phases with the concentrations (n−1)/9 and n/9 can be represented by the expression
(8)$$\begin{array}{cc} \; & \mu_{k,l}=3k+3\left(l-1\right)J_{2,5},\;k=0,1,2;\;l=0,1,2,\\ \; & n=1+3k+l. \end{array}$$

The phase diagram of the system for the larger interaction parameter, $J_{2,5}>1/3$, is shown in Figure 6 as well. In this case, the phase coexistence lines between the phases with concentrations (n−1)/9 and n/9 obey the expression
(9)$$\begin{array}{cc} \; & \mu_{k,l}=k+2l+3\left(l-1\right)J_{2,5},\;k=0,1,2;\;l=0,1,2,\\ \; & n=1+k+3l. \end{array}$$

The dependence of the potential ω on the chemical potential at $J_{2,5}>1/3$ is shown in Figure 7. The stable states correspond to the lowest line segments between the intersection points, which indicate the chemical potential values for the phase coexistence.

The structure of the stable phases at $J_{2,5}>1/3$ is shown in Figure 8. At the concentrations 3/9, 4/9, 5/9, and 6/9, the degenerated ground states exist. E.g., either triangles of the nearest neighbors or stripes parallel to the lattice vectors can exist with equal probabilities at c=3/9. Ordered rhombuses or stripes with additional particles attached to them can occur at the concentration c=4/9.

At the crossover value of the parameter $J_{2,5}=1/3$, all the system states for the concentration between 2/9 and 7/9 are degenerated. The $\omega_{n/9}$ functions of the chemical potential μ according to Equations (Equation 3)–(Equation 7) coincide for all possible system states for a given value of *n*. Any distribution of particles shown in columns a) and c) of Figure 3 can occur at this value of the parameter $J_{2,5}$ on the line segments between the intersection points at μ=n−2 for the coexisting phases c=(n−1)/9 and c=n/9,n=1,2,...,9.

### 3.2. The Ground States for Fixed Number of Particles

In systems with a fixed number of particles, an arbitrary mean concentration can be considered. At c≠n/9, two phases separated by an interface line coexist. Like in the previous case [15], the interface lines can be parallel or perpendicular to the lattice vectors $e_{i},i=1,2,3$ (Figure 3). We have verified that the interface lines parallel to the lattice vectors are preferable because their line tensions are smaller.

On the coexistence line between vacuum and the c=1/9 phase, each particle of the latter phase loses two interacting bonds with the fifth neighbors in the vacuum phase. The distance between particles along the interface is equal 3a, where *a* is the nearest neighbor distance. Thus, the line tension is equal to $\sigma =J_{2,5}/3a$. At the interface line perpendicular to the lattice vectors $e_{i}$, each particle in the first row loses three interacting bonds and in the second row one interacting bond. The distance between particles along the interface line is equal to $3a\sqrt{3}$. The line tension is $2J_{2,5}/3a\sqrt{3}>J_{2,5}/3a$. Thus, the interface lines between the coexisting phases c=0 and c=1/9 are parallel to the lattice vectors. The same conclusion, $\sigma =J_{2,5}/3a$, follows for the interface line between the phases c=1/9 and c=2/9 in both cases $J_{2,5}<1/3$ and $J_{2,5}>1/3$. Figure 9 demonstrates that one of the particles in the near interface unit cell of the c=2/9 phase has six neighbors of the fifth order, while the other one has four such neighbors. The minimum of the line tension is assumed when one of the particles of the c=2/9 phase in the unit cell has two fifth neighbors and no first and second neighbors in the phase c=1/9. We finally note that the interface is twofold degenerated for the case of $J_{2,5}>1/3$ at the concentrations 3/9≤c≤6/9 (see Figure 8).

As an example of a structure in a system with a fixed number of particles, the simulation snapshot for the system of 37 particles on the lattice of 36 × 36 lattice sites (the concentration c=0.029) is shown in Figure 10. In the ideal case of T=0, the particles have to form a regular hexagon. However, the simulation was done at a quite low but nonzero temperature T=0.1. As a result, the coexistence of the rarefied gas phase (the vacuum state with a few evaporated particles as defects) and the phase with c=1/9 is obtained. The interface lines are parallel to the lattice vectors in agreement with the analytical calculation for T=0.

At $J_{2,5}<1/3$, the system states for a subsequent concentration can be produced from the previous one by adding a particle in the unit cell. At the interface line, this particle has no counterparts in the lower concentration phase, losing two interacting bonds with the fifth neighbors and the line tension is equal $\sigma =J_{2,5}/3a$ for all the coexisting phases with concentrations (n−1)/9 and n/9 for n=1,2,...,9.

The situation at $J_{2,5}>1/3$ is more complicated. The neighboring system states in the upper row in Figure 3 differ from each other by one particle in the unit cell and the line tension for all the interfaces parallel to the lattice vectors is equal to $J_{2,5}/3a$. The system state for the c=3/9 phase in the lower row can coexist with the c=2/9 phase, because this state differs from the previous phase by one particle as well. Thus, the degeneracy of the phase coexistence between the phases c=2/9 and c=3/9 is observed.

However, the c=4/9 phase in the lower row in Figure 3 can coexist with the c=3/9 phase of the upper row, but not with that of the lower row, because they differ by positions of more than one particle. Thus, three combinations of coexisting phases exist at the mean concentration 3/9<c<4/9. The same situation exists for the coexistence of the c=4/9 and c=5/9 phases. The states of the upper row can coexist between themselves as well as with the cross phases of the lower row. However, these phases in the lower row cannot form a stable interface between themselves. There are no counterparts for two particles in the more concentrated phase as well as for one particle of the less concentrated phase. The line tension in this case is three times larger. Three combinations of coexisting phases separated by a stable interface exist at the mean concentration 4/9<c<5/9 as well.

The system states are symmetric with respect to μ=3 or c=0.5 and the particle-vacancy interchange [18]. The phase coexistence at larger chemical potentials and concentrations are symmetric to their lower values.

## 4. Thermodynamics of the System at T>0

At low dimensionless temperatures $T=k_{B}T^{*}/J$ (where $T^{*}$ is the absolute temperature and $k_{B}$ the Boltzmann constant), the ordered states found in the ground state remain present in the system, while the ordering is destroyed gradually with the temperature increase, due to thermal fluctuations. In this section, the Monte Carlo (MC) simulation results for μ(c) isotherms, isothermal compressibility, and specific heat are presented for the system at two values of the interaction parameter $J_{2,5}=1/4$ and $J_{2,5}=1/2$ below and above the crossover value $J_{2,5}=1/3$. The Metropolis importance sampling simulations were performed with the chemical potential step Δμ=0.02 for the system of 96×96 lattice sites with periodic boundary conditions. one thousand Monte Carlo simulation steps (MCS) were used for equilibration. The subsequent 10 000 MCS were used for calculating the average values.

The isotherms displaying the concentration dependence on the chemical potential at $J_{2,5}=1/2$ demonstrate typical behavior at low temperatures T=0.1, 0.2, and 0.3 (Figure 11). The wide empty horizontal segments in the μ(c) plots indicate forbidden regions of concentrations. These two-phase segments are separated by very steep parts of the μ(c) plots, where μ increases rapidly for very narrow range of *c*, centered at n/9. These intervals of *c* separating the horizontal segments expand with increasing temperature. The concentration intervals around n/9,n=0,1,...,9 correspond to the ordered patterns discussed in the previous section in the case of the grand canonical ensemble. At larger temperature T=0.5, the horizontal regions in the μ(c) plot almost disappear. Thus, the critical temperature can be estimated as $T_{cr}≃0.6$. The concentration increases continuously as a function of the chemical potential at this temperature. The repulsion interaction between the second neighbors not only removes the degeneracy of the system states at the concentrations 1/3 and 2/3, but also significantly reduces the critical temperature, which is around 1.1 when the second neighbors do not interact [15].

The structure of stable phases resembles the ground state configurations. In Figure 12, Figure 13 and Figure 14 the snapshots of the system at the concentration close to 4/9 are shown as examples. At the interaction parameter $J_{2,5}>1/2$ above its critical value 1/3, the system is degenerated and in different runs the final state Figure 13 and Figure 14 corresponds to the possible ground state configurations given in Figure 8 with a few defects, due to thermal fluctuations.

The phase transitions can be more clearly revealed by considering fluctuations (Figure 15). The inverse value of the thermodynamic factor $\chi_{T}=c{\left(\partial \left(\beta \mu \right)/\partial c\right)}_{T}$ is proportional to the concentration fluctuations
(10)$$\chi_{T}^{-1}=\frac{\langle {\left(N-\langle N\rangle \right)}^{2}\rangle }{\langle N\rangle }$$
that in turn is proportional to the isothermal compressibility $\kappa_{T}={\left(\partial c/\partial p\right)}_{T}/c$, $\chi_{T}^{-1}=cT\kappa_{T}$, where the angular brackets 〈...〉 denote averaging over the grand canonical ensemble, *N* is the number of particles in the system, c=〈N〉/M is the mean lattice concentration, and *p* is the pressure.

At the lowest temperatures, T=0.1 and T=0.2, the concentration fluctuations of each phase exist in narrow concentration regions. At the temperature T=0.4, the minima of the concentration fluctuations are well distinguishable. They are attained at the most ordered system states with concentrations equal to a multiple 1/9. At T=0.5, the concentration fluctuations span almost over the entire concentration range (0,1), indicating the approach to the critical temperature.

Similar behavior is observed for the dimensionless specific heat (Figure 16), which is proportional to the energy fluctuations
(11)$$c_{V}=\frac{1}{k_{B}\langle N\rangle }{\left(\frac{\partial E^{*}}{\partial T^{*}}\right)}_{\mu }=\frac{\langle {\left(E-\langle E\rangle \right)}^{2}\rangle }{\langle N\rangle T^{2}},$$
where $E=E^{*}/J$ is the dimensionless system energy (see the first term on the RHS in Equation (Equation 1)), and
(12)$$\langle E\rangle =\frac{1}{2}\langle \sum_{k=1}^{5}\sum_{k_{i}=1}^{z_{k}}\sum_{i=1}^{M}J_{k}{\hat{n}}_{i}{\hat{n}}_{k_{i}}\rangle .$$

As an example, the fine structure of the concentration and energy fluctuations is shown in Figure 17 and simulated with the reduced chemical potential step. The minima of these characteristics are close to the concentration c=1/3 of the most ordered system state.

Structural peculiarities of the system can be tracked by considering the order parameters. The occupancy of particular sublattices (Figure 2) represents nine such order parameters. At the lowest temperatures, T=0.1 and 0.2, the succession of the order parameters is in fact represented by the step functions rising from 0 to 1 when the concentration attains the value equal to a multiple of 1/9. The order parameters become smoother when the temperature increases.

For the interaction parameter below its critical value 1/3, in particular for $J_{2,5}=1/4$, the chemical potential isotherms look like in the previous case (Figure 18), but the critical temperature is even lower, around 0.4. At low temperatures, the inverse thermodynamic factor and the specific heat, as well as the order parameters, have the same prominent features at the concentrations around a multiple of 1/9. The rapid changes of the above quantities for c≈n/9 are smoothed out with the temperature increase.

Despite very similar thermodynamic properties of the systems with the interaction parameter $J_{2,5}$ above and below its crossover value $J_{2,5}=1/3$, the structural characteristics are completely different above and below the crossover. The particle distribution over the lattice sites corresponds to the structures of Figure 5 or Figure 8 in the former and in the latter case, respectively. Due to thermal fluctuations at non-zero temperatures, the structures contain defects, which are particles on the sites that do not belonging to the ideal configurations or vacant sites on these configurations. The number of defects increases with the temperature increase, and the concentration range corresponding to the ordered phases increases as well.

## 5. Conclusions

We considered pattern formation by particles with hard cores covered by thick polymeric shells on a fluid interface. The structure of the shell can be designed in experimental studies, and it determines the effective repulsion between the particles. For this reason, the aim of our study was the determination of the effect of the shape of the repulsive potential on the patterns formed by the particles. We focused on the question how the rate at which the repulsion decreases with the distance influences the pattern formation. We considered a triangular lattice model with the lattice sites occupied by the particle cores. The nearest and next nearest neighbors repel each other, due to the overlapping polymeric shells, while the fifth neighbors attract each other because of the capillary forces.

The second neighbor repulsion in addition to the nearest and fifth neighbor interaction results in the significant enrichment of the ground state structures. Alongside with the concentrations equal to a multiple of 1/3, the states with the concentrations equal to a multiple of 1/9 are present for certain intervals of the chemical potential.

Importantly, the patterns formed for the concentration c=n/9,n=2,...,7, are completely different for the repulsion that shows a fast and a slow decrease with the interparticle separation (see Figure 5 and Figure 8 for the first and the second case, respectively). In the second case, two quite different patterns can occur with the same probability for given μ. The crossover value of the second-neighbor repulsion (equal to the fifth neighbor attraction) separating the two types of patterns is $J_{2,5}=1/3$. Our model is suitable for thick composite shells, with a stiff inner part and soft outer part. The results show that by modifying the thickness and the structure of the stiff inner part, we can obtain completely different patterns on an interface.

In systems with a fixed number of particles, the energetically preferable interface lines are parallel to the lattice vectors for all stable interfaces. We have shown that in the case of the degenerated ground states, two types of patterns with lower density can coexist with two patterns with higher density, giving together four pairs of coexisting patterns. A stable interface, however, cannot be formed for one of these pairs.

At non-zero but not too large temperatures, the system passes through the structures corresponding to the ground states with thermally initiated defects. At low temperatures, the stable states exist in very narrow concentration intervals close to the concentrations equal to a multiple of 1/9. The concentration intervals enlarge with the temperature increase. At temperatures slightly above the critical one, the concentration versus chemical potential isotherms become continuous. The critical temperature depends strongly on the shape of the interaction potential too. In the case of $J_{2,5}=1/2$, i.e., when the intensity of the second neighbor repulsion is equal to the fifth neighbor attraction and twice as low as the first neighbor repulsion, the critical temperature is almost two times lower than in the system with vanishing interaction between the second neighbors [15]. The critical temperature decreases with decreasing interaction between the second and the fifth neighbors.

The fluctuations of the number of particles and energy are maximal at the concentrations corresponding to the phase transition points, and minimal in the most ordered states at concentrations close to a multiple of 1/9. The order parameters determined as the mean concentrations on the sublattices demonstrate fast increase from 0 to 1, while the mean system concentration crosses a value equal to a multiple of 1/9.

Quantitative comparison of the predictions of our model with experimental results is not possible yet. Due to the sensitive dependence of the patterns formed by the particles on the interfaces on the details of the effective interactions, it is necessary to precisely design and control the crosslinking architecture of the polymeric shell, in order to obtain the desired shape of the effective interactions in experiment. On the theoretical side, it would be necessary to compute the pressure–area fraction isotherms that are measured in experiments. On the qualitative level, however, our predictions for the chemical potential—area fraction isotherms can be compared with the experimental pressure–area fraction isotherms for the CSNPs with the shell-to-core size ratio ∼3, since the chemical potential and the pressure play similar roles. Indeed, steps in the isotherms are clearly seen in Ref. [3], in qualitative agreement with our results. Moreover, structural evolution for increasing area fraction, in particular the formation of clusters, and the orientation of the interface between coexisting phases in our theory and in experiment are similar.

To summarize, we stress that our model indicates that by careful construction of the polymeric shell, one should be able to obtain core-shell nanoparticles forming a variety of different patterns on an interface. 

## Figures and Tables

**Figure 1 entropy-22-01215-f001:**
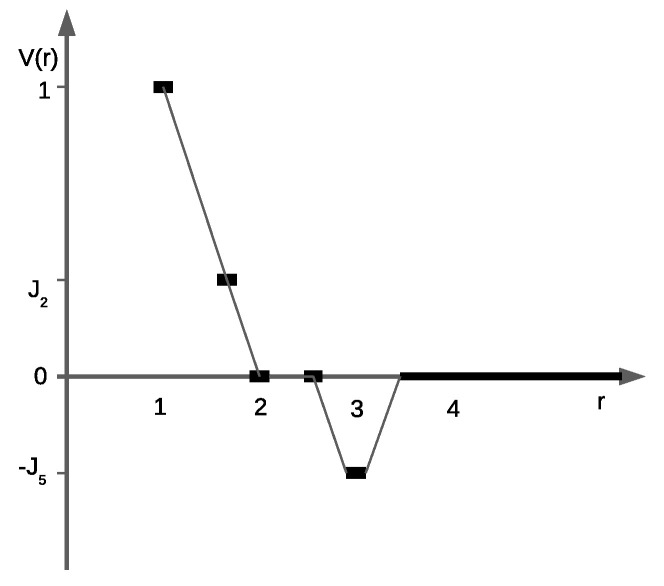
The interaction potential as a function of the distance between the particle centers in units of the first-neighbor repulsion *J*. The symbols denote the interaction between the cores occupying the lattice sites, and the line is to guide the eye. The shown interactions $J_{2}=J_{5}=1/3$ correspond to a crossover between different patterns formed by the particles, as described in Section 3.1.

**Figure 2 entropy-22-01215-f002:**
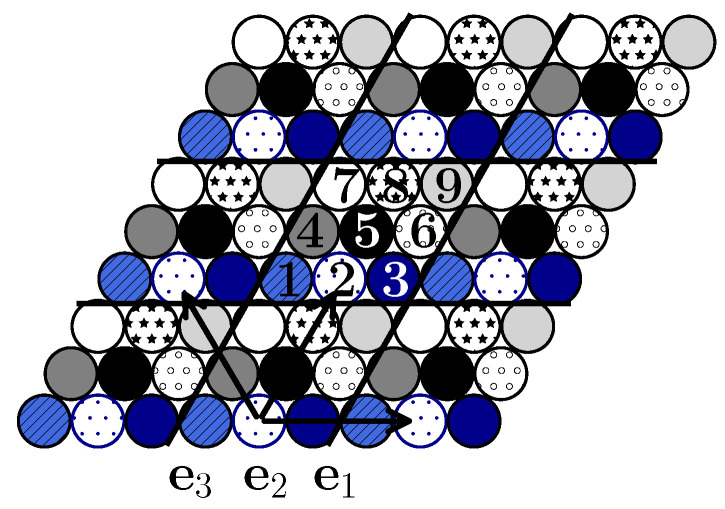
The system of unit cells with particles belonging to nine sublattices and the lattice vectors $e_{i},i=1,2,3$. The particles 1 and 2 or 1 and 4 are the nearest neighbors, the particles 1 and 5 or 2 and 7 are the next nearest neighbors, the particles 1 and 3 or 1 and 7 are the third neighbors, the particles 1 and 6 are the fourth neighbors. The particles with the same texture in nearest unit cells with the separation 3a are the fifth neighbors.

**Figure 3 entropy-22-01215-f003:**
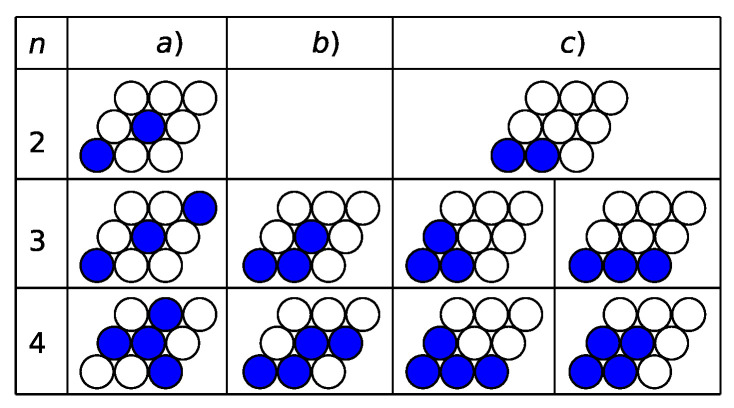
The possible distributions of particles over the unit cell for the concentrations c=n/9 with 2≤n≤4. For the concentrations with 5≤n≤7, the particles and vacancies have to be interchanged. For the concentrations with n=1 or 8, the particle or vacancy can occupy any lattice site of the unit cell. There are several equivalent distributions of particles over the unit cell for the concentrations with 2≤n≤4 or vacancies for the concentrations with 5≤n≤7.

**Figure 4 entropy-22-01215-f004:**
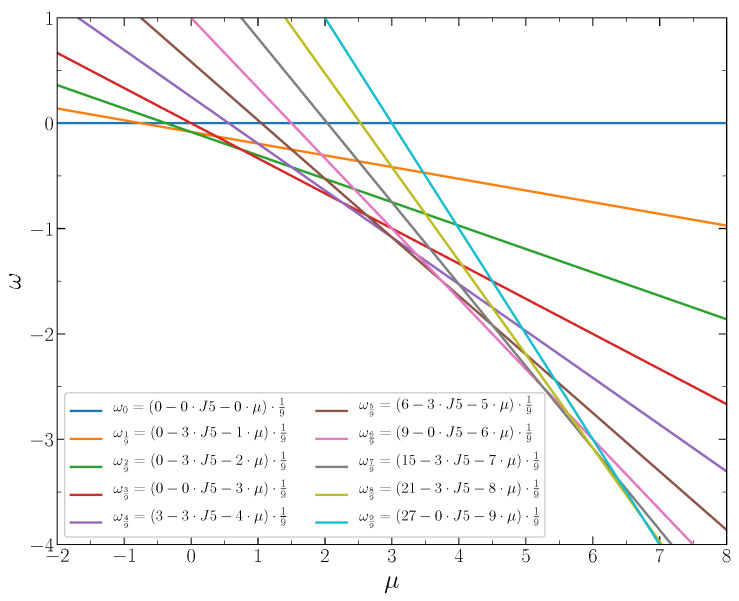
The dimensionless thermodynamic Hamiltonian per lattice site versus the chemical potential for the concentrations c=n/9,n=0,1,2,...9 at $J_{2,5}=1/4$.

**Figure 5 entropy-22-01215-f005:**
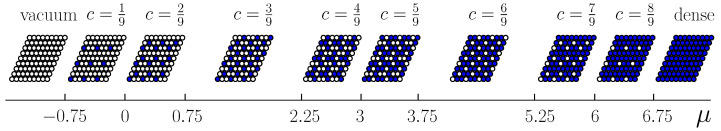
Cartoons of the distribution of particles over the lattice sites and the corresponding chemical potential intervals for the system with $J_{2,5}=1/4$.

**Figure 6 entropy-22-01215-f006:**
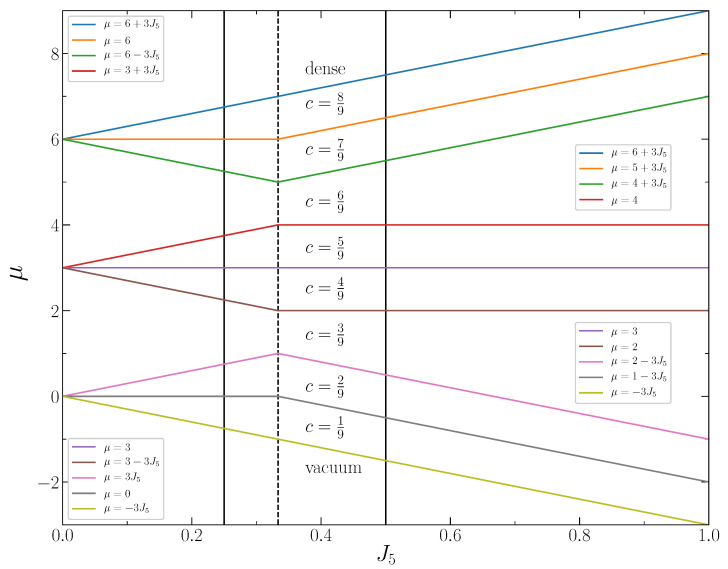
The phase diagram of the system. The vertical lines show the chemical potential intervals for the system states at $J_{2,5}=1/4$, $J_{2,5}=1/3$ and $J_{2,5}=1/2$.

**Figure 7 entropy-22-01215-f007:**
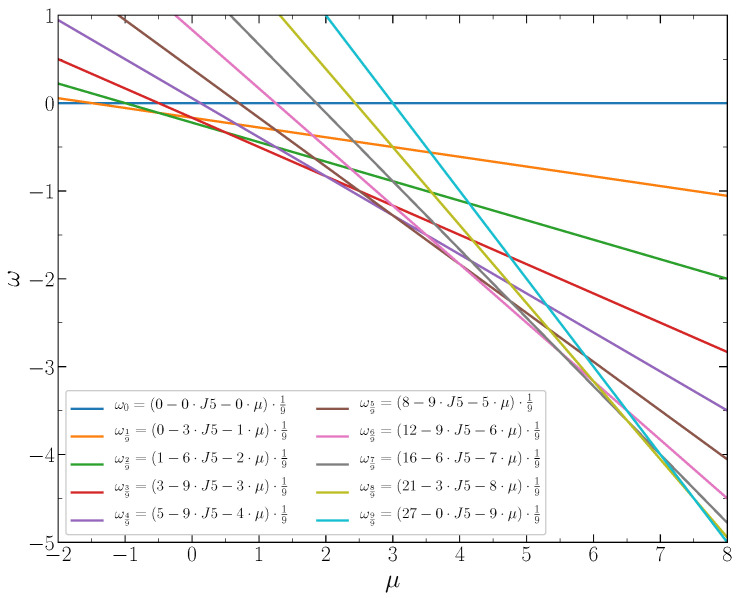
The dimensionless thermodynamic Hamiltonian per lattice site versus the chemical potential for the concentrations c=n/9,n=0,1,2,...,9 at $J_{2,5}=1/2$.

**Figure 8 entropy-22-01215-f008:**
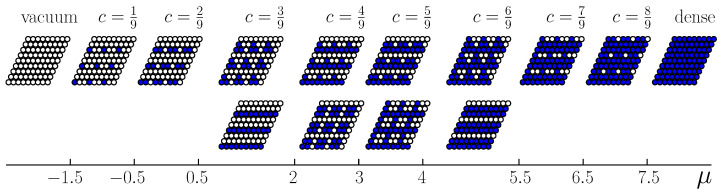
Cartoons of the distribution of particles over the lattice sites and the corresponding chemical potential intervals for the system with $J_{2,5}>1/3$.

**Figure 9 entropy-22-01215-f009:**
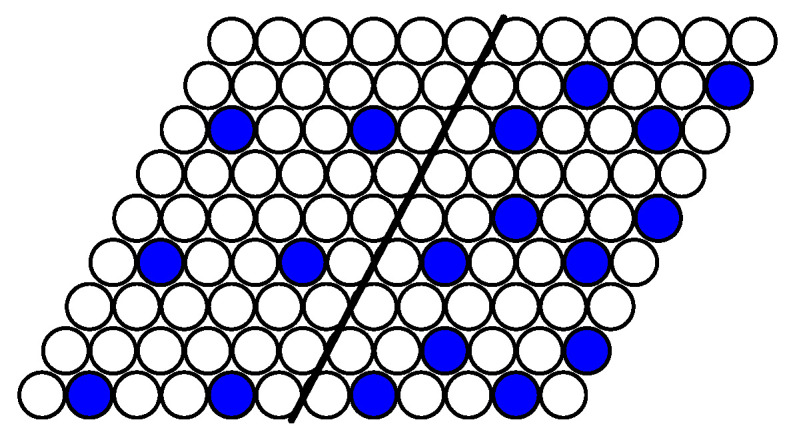
Cartoon of the interface between the c=1/9 and 2/9 phases for the system at $J_{2,5}<1/3$.

**Figure 10 entropy-22-01215-f010:**
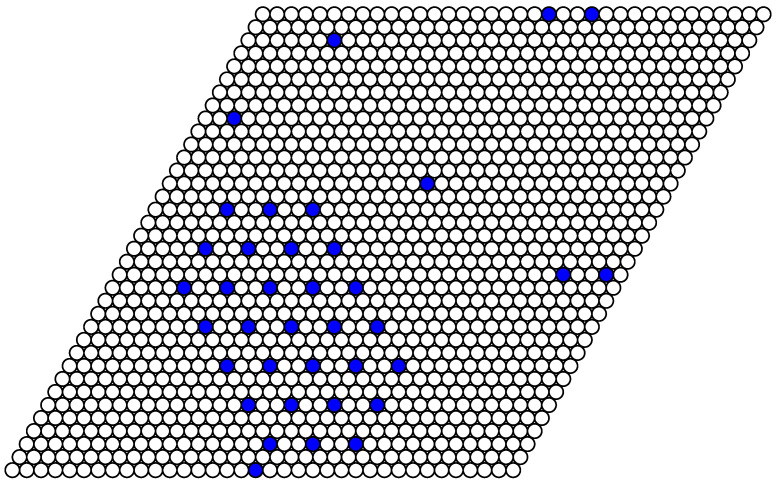
The snapshot of the system of 37 particles on the lattice of 36 × 36 lattice sites (c≃0.029) after 9000 Monte Carlo simulation steps (MCS) at $J_{2,5}=1/2$. The interface lines are parallel to the lattice vectors $e_{i}$.

**Figure 11 entropy-22-01215-f011:**
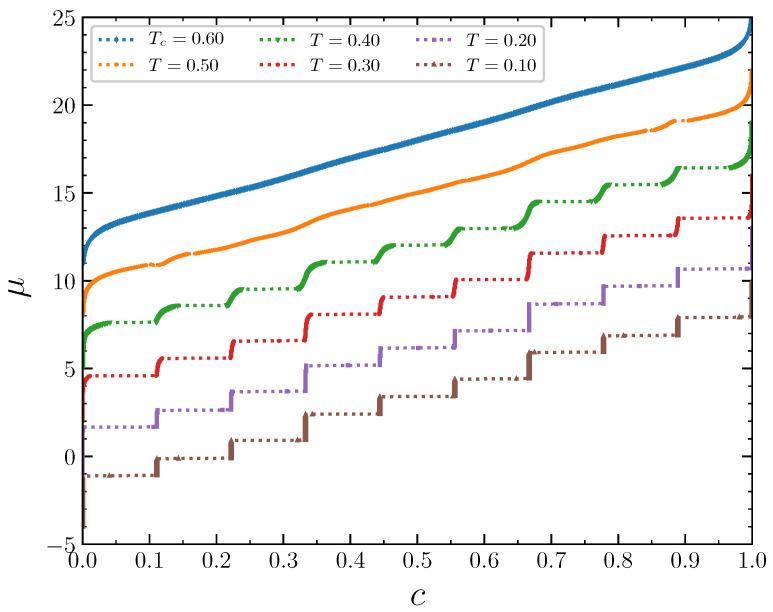
The chemical potential as a function of the concentration at $J_{2,5}=0.5$ and several temperatures. The isotherms are shifted in the vertical direction by 3 from each other for clarity. The isotherm at T=0.1 is not shifted.

**Figure 12 entropy-22-01215-f012:**
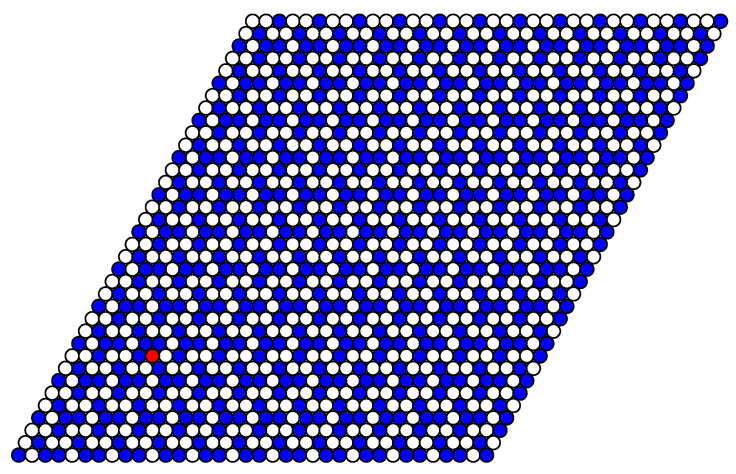
Snapshot of the system at T=0.2, μ=2.6, $J_{2,5}=1/4$ after 8000 MCS. The extra particle (defect) is shown in red. This structure corresponds to the ground state configuration shown in Figure 5.

**Figure 13 entropy-22-01215-f013:**
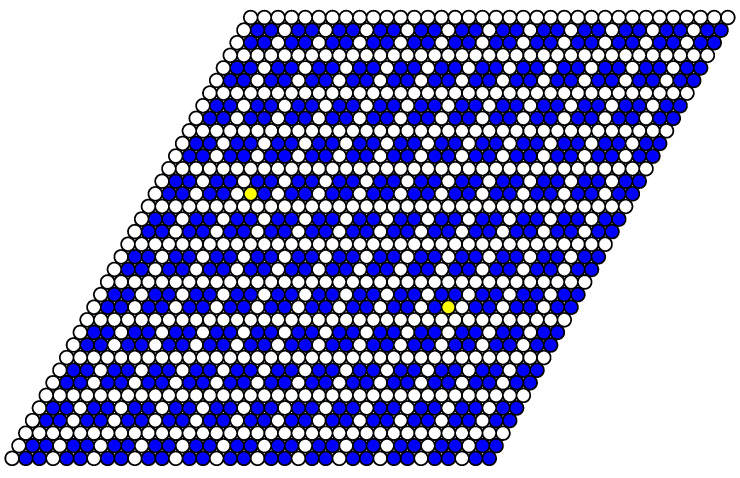
Snapshot of the system at T=0.3, μ=2.6, $J_{2,5}=1/2$ after 8000 MCS. The additional vacancies (defects) are shown in yellow. This structure corresponds to the ground state configuration shown in the bottom row of Figure 8.

**Figure 14 entropy-22-01215-f014:**
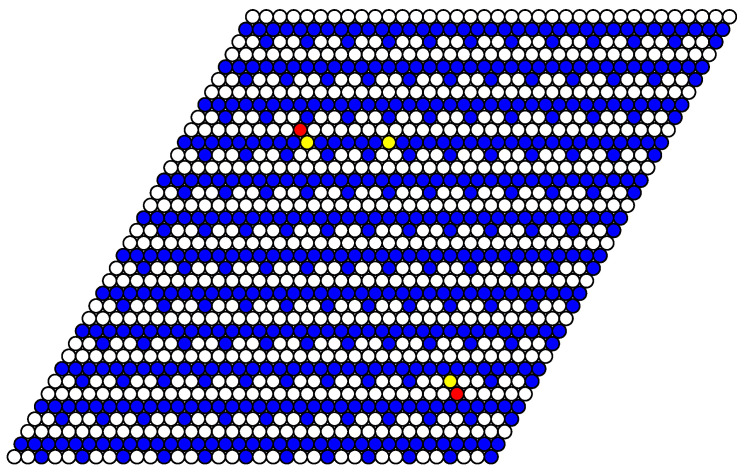
Snapshot of the system at T=0.3, μ=2.6, $J_{2,5}=1/2$ after 8000 MCS. The extra particles and additional vacancies (defects) are shown in red and yellow, respectively. This structure corresponds to the ground state configuration shown in the upper row of Figure 8.

**Figure 15 entropy-22-01215-f015:**
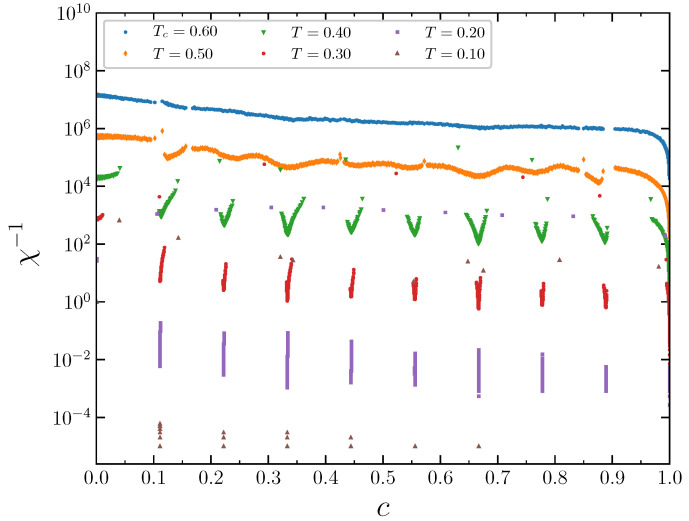
The inverse thermodynamic factor as a function of the concentration at $J_{2,5}=0.5$ and several temperatures. The curves are shifted in the vertical direction by $3^{3n}$ from the lowest one for clarity.

**Figure 16 entropy-22-01215-f016:**
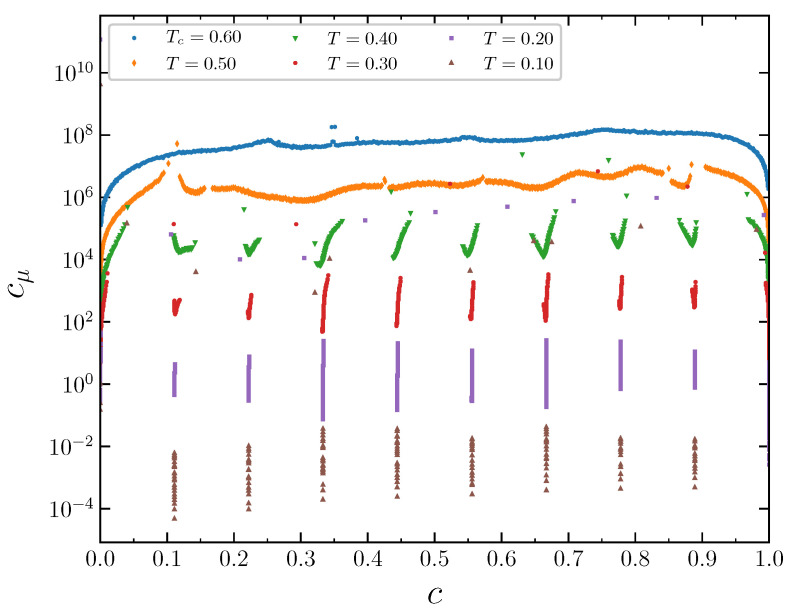
The dimensionless specific heat as a function of the concentration at $J_{2,5}=0.5$ and several temperatures. The curves are shifted in the vertical direction by $3^{3n}$ from the lowest one for clarity.

**Figure 17 entropy-22-01215-f017:**
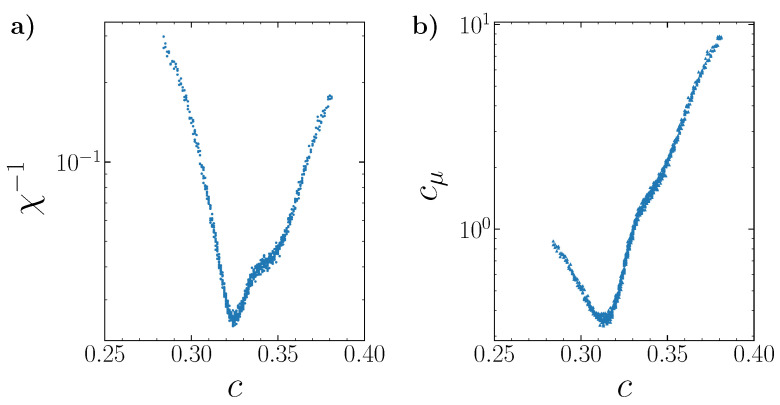
The fine structure of the inverse thermodynamic factor (**a**) and dimensionless specific heat (**b**) at T=0.3 and $J_{2,5}=0.5$ simulated with the reduced chemical potential step Δμ=0.002. The scatter of the results characterizes the precision of the simulation.

**Figure 18 entropy-22-01215-f018:**
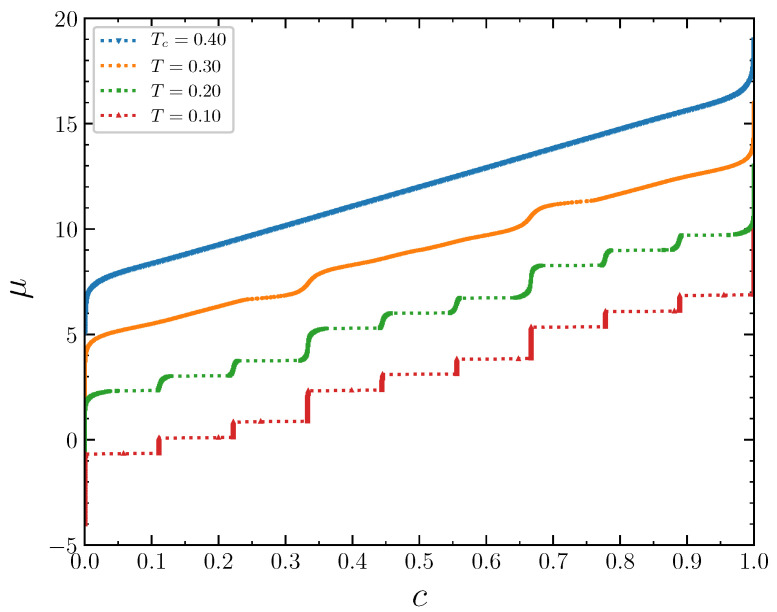
The chemical potential as a function of the concentration at $J_{2,5}=0.25$ and several temperatures. The isotherms are shifted in the vertical direction by 3 from each other for clarity. The isotherm at T=0.1 is not shifted.

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
