# Peer review of "Structural and Thermodynamic Peculiarities of Core-Shell Particles at Fluid Interfaces from Triangular Lattice Models"

_entropy, 2020, doi:10.3390/e22111215_

Round 1

Reviewer 1 Report

See the attached.

Reviewer 2 Report

The authors present results for a 2D lattice model inspired by core-shell nanoparticles (NPs) at an interface.  The potential energy between sites incorporates two-scale repulsion (driven shell-shell and core-core interactions) and longer range attraction due to capillary forces. The authors calculate thermodynamic quantities for both open and closed ensembles at T=0, varying key interactions parameters, and carry out MC simulations at finite T, reporting phase diagrams and data on thermodynamic response functions, where the ordered phases are based on a 9-site unit cell.  The work builds on previous 1-D and 2-D modelling of the system.

I have not checked details of the calculations.

The work should be of interest to both researchers interested in understanding the phase behaviour of NPs at interfaces, and those interested in lattice models inter own right, perhaps for pedagogical reasons.  I recommend publication.

Minor points

line 116-177 "J2 and J5 are dimensionless ... energies in unit of ... J"  This sentence appears contradictory.  J2 and J5 are dimensionless.  J has units of energy.  Please reword. 

line 124.  Please explain in detail why only including up to the 5th nn interactions implies that only the 9-site unit cell needs to be considered. 

line 176  looses -> loses
